# DIFFERENTIABLE TOP-$k$ CLASSIFICATION LEARNING

## ABSTRACT

The top-$k$ classification accuracy is one of the core metrics in machine learning. Here, $k$ is conventionally a positive integer, such as $1$ or $5$. In this work, we relax this assumption and propose to draw $k$ from a probability distribution for training. Combining this with recent advances in differentiable sorting and ranking, we propose a new family of differentiable top-$k$ cross-entropy classification losses. We find that relaxing $k$ does not only produce better top-5 accuracies, but also makes models more robust, which leads to top-1 accuracy improvements. When fine-tuning publicly available ImageNet models, we achieve a new state-of-the-art on ImageNet for publicly available models with an $88.37\%$ top-1 and a $98.68\%$ top-5 accuracy.

## 1 INTRODUCTION

Classification is one of the core disciplines in machine learning and computer vision. With the advent of classification problems with hundreds or even thousands of classes, the top-$k$ classification accuracy has established itself as an important task, i.e., an algorithm can suggest $k$ classes and one of them has to be the correct class. Usually, models are trained to optimize the top-1 accuracy and top-5 etc. are used for evaluation only. Some works (Lapin et al., 2016; Berrada et al., 2018) have challenged this idea and proposed top-$k$ losses, such as a smooth top-5 margin loss. These methods have demonstrated superior robustness over the established top-1 softmax cross-entropy in presence of additional label noise (Berrada et al., 2018). In standard classification settings, however, these methods have so far not shown improvements over the established top-1 softmax cross-entropy.

In this work, instead of selecting a single top-$k$ metric such as top-1 or top-5 for defining the loss, we propose to specify $k$ to be drawn from a probability distribution $P_K$, which may or may not depend on the confidence of specific data points or on the class label. Examples for distributions $P_K$ are $[.5, 0, 0, 0, .5]$ (50% top-1 and 50% top-5), $[.1, 0, 0, 0, .9]$ (10% top-1 and 90% top-5), and $[.2, .2, .2, .2, .2]$ (20% top-$k$ for each $k$ from 1 to 5). Note that, when $k$ is drawn from a distribution, this is done sampling-free as we can compute the expectation value in closed form.

Conventionally, given scores returned by a neural network, softmax produces a probability distribution over the top-1 rank. Recent advances in differentiable sorting and ranking (Grover et al., 2019; Prillo & Eisenschlos, 2020; Cuturi et al., 2019; Petersen et al., 2021) provide methods for generalizing this to probability distributions over all ranks represented by a matrix $\boldsymbol{P}$. Based on differentiable ranking, multiple differentiable top-$k$ operators have recently been proposed. They found applications in differentiable $k$-nearest neighbor, differentiable beam search, attention mechanisms, and differentiable image patch selection (Cordonnier et al., 2021). In these areas, integrating differentiable top-$k$ improved results considerably by creating a more natural end-to-end learning setting. However, to date, none of the differentiable top-$k$ operators have been employed as neural network losses for top-$k$ classification learning with $k > 1$.

Building on differentiable sorting and ranking methods, we propose a new family of differentiable top-$k$ classification losses where $k$ is drawn from a probability distribution. We find that our top-$k$ losses improve not only top-$k$ accuracies, but also top-1 accuracy on multiple learning tasks.

We empirically evaluate our method using four differentiable sorting and ranking methods on the CIFAR-100 (Krizhevsky et al., 2009), ImageNet-1K (Deng et al., 2009), and the ImageNet-21K-P (Ridnik et al., 2021) data sets. Using CIFAR-100, we demonstrate the capabilities of our losses to train models from scratch. On ImageNet-1K, we demonstrate that our losses are capable of fine-tuning state-of-the-art models and achieve a new state-of-the-art for publicly available mod-

els on both top-1 and top-5 accuracy. We benchmark our method on multiple recent models and demonstrate that our proposed method consistently outperforms the baselines for the best two differentiable sorting and ranking methods. With ImageNet-21K-P, where many classes overlap (but only one is the ground truth), we demonstrate that our losses are scalable to more than $10\,000$ classes and achieve improvements of over $1\%$ with only last layer fine-tuning.

Overall, while the performance improvements on fine-tuning are rather limited (because we retrain only the classification head), they are consistent and can be achieved without the large cost of training from scratch. The absolute $0.2\%$ improvement that we achieve on the ResNeXt-101 32x48d WSL top-5 accuracy corresponds to an error reduction by approximately $10\%$, and can be achieved at much less than the computational cost of (re-)training the full model in the first place.

We summarize our contributions as follows:

- We derive a novel family of top-$k$ cross-entropy losses and relax the assumption of a fixed $k$.
- We find that they improve both top-$k$ and top-1 accuracy.
- We demonstrate its scalability to more than $10\,000$ classes.
- We propose splitter selection nets, which require fewer layers than existing selection nets.
- We achieve new state-of-the-art results (for publicly available models) on ImageNet1K.

## 2 BACKGROUND: DIFFERENTIABLE SORTING AND RANKING

We briefly review NeuralSort, SoftSort, Optimal Transport Sort, and Differentiable Sorting Networks. We omit the fast differentiable sorting and ranking method (Blondel et al., 2020b) as it does not provide relaxed permutation matrices / probability scores, but rather only sorted / ranked vectors.

### 2.1 NEURALSORT & SOFTSORT

To make the sorting operation differentiable, Grover et al. (2019) proposed relaxing permutation matrices to unimodal row-stochastic matrices. For this, they use the softmax of pairwise differences of (cumulative) sums of the top elements. They prove that this, for the temperature parameter approaching $0$, is the correct permutation matrix, and propose a variety of deep learning differentiable sorting benchmark tasks. They propose a deterministic softmax based variant, as well as a Gumbel-Softmax variant of their algorithm. Note that NeuralSort is not based on sorting networks.

Prillo & Eisenschlos (2020) build on this idea but simplify the formulation and provide SoftSort, a faster alternative to NeuralSort. They show that it is sufficient to build on pairwise differences of elements of the vectors to be sorted instead of the cumulative sums. They find that SoftSort performs approximately equivalent in their experiments to NeuralSort.

### 2.2 OPTIMAL TRANSPORT / SINKHORN SORT

Cuturi et al. (2019) propose an entropy regularized optimal transport formulation of the sorting operation. They solve this by applying the Sinkhorn algorithm (Cuturi, 2013) and produce gradients via automatic differentiation rather than the implicit function theorem, which resolves the need of solving a linear equation system. As the Sinkhorn algorithm produces a relaxed permutation matrix, we can also apply the Sinkhorn sort to top-$k$ classification learning.

### 2.3 DIFFERENTIABLE SORTING NETWORKS

Petersen et al. (2021) propose differentiable sorting networks, a continuous relaxation of sorting networks. Sorting networks are a kind of sorting algorithm that consist of wires carrying the values and comparators, which swap the values on two wires if they are not in the desired order. Sorting networks can be made differentiable by perturbing the values on the wires of the sorting network in each layer of the sorting network by a logistic distribution, i.e., instead of $\min$ and $\max$ they use softmin and softmax. Similar to the methods above, this method produces a relaxed permutation matrix, which allows us to apply it to top-$k$ classification learning. Note that sorting networks are a classic algorithmic concept (Knuth, 1998a) are not neural networks nor refer to differentiable sorting. Differentiable sorting networks are one of multiple differentiable sorting and ranking methods.

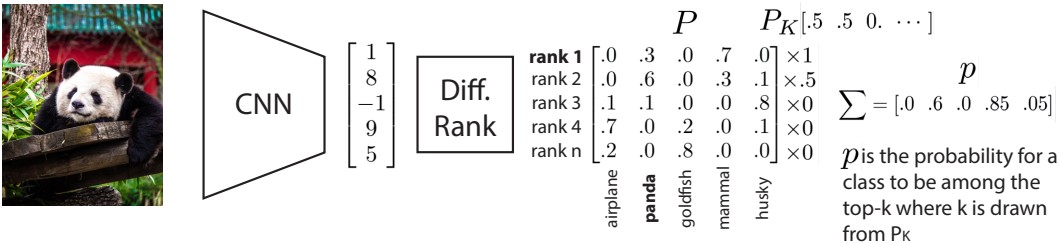

Figure 1: Overview of the proposed architecture: A CNN predicts scores for an image, which are then ranked by a differentiable ranking algorithm returning the probability distribution for each rank in matrix $\boldsymbol{P}$. The rows of this distribution correspond to ranks, and the columns correspond to the respective classes. In the example, we use a $50\%$ top-1 and $50\%$ top-2 loss, i.e., $P_K = [.5, .5, 0, 0, 0]$. Here, the $k$th value refers to the top-$k$ component, which is satisfied if the prediction is at *any* of rank-1 to rank-$k$. Thus, the weights for the different ranks can be computed via a cumulative sum and are $[1, .5, 0, 0, 0]$. The correspondingly weighted sum of rows of $\boldsymbol{P}$ yields the probability distribution $p$, which can then be used in a cross-entropy loss. Photo by Chris Curry on Unsplash.

## 3 TOP-$k$ LEARNING

In this section, we start by introducing our objective, elaborate its exact formulation, and then build on differentiable sorting principles to efficiently approximate the objective. A visual overview over the loss architecture is also given in Figure 1.

The goal of top-$k$ learning is to extend the learning criterion from only accepting exact (top-1) predictions to accepting $k$ predictions among which the correct class has to be. In its general form, for top-$k$ learning, $k$ may differ for each application, class, data point, or a combination thereof. For example, one practitioner may want to rank 5 predictions and assign a score that depends on the rank of the true class among these ranked predictions, while, on the other hand, another practitioner may want to obtain 5 predictions but does not care about their order. Yet another practitioner in image classification may want to enforce a top-1 accuracy on images from the "person" super-class, but resign to a top-3 accuracy for the "animal" super-class, as it may have more ambiguities in class-labels. We model this by a random variable $K$, following a distribution $P_K$ that describes the relative importance of different values $k$. The discrete distribution $P_K$ is either a marginalized distribution for a given setting (such as the uniform distribution), or a conditional distribution for each class, data point, etc. This allows the practitioner to specify a marginalized / conditional distribution $k \sim P_K$. This generalizes the ideas of conventional top-1 supervision (usually softmax cross-entropy) and top-$k$ supervision for a $k$ like $k = 5$ (usually based on surrogate top-$k$ margin/hinge losses like (Lapin et al., 2016; Berrada et al., 2018)) and unifies them.

The objective of top-$k$ learning is maximizing the probability of accepted predictions of the model $f_\Theta$ on data $X, y \sim \mathcal{D}$ given marginal distribution $P_K$ (or conditional $P_{K|X,y}$ if it depends on the class $y$ and/or data point $X$). In the following, $\boldsymbol{P}_{k,y}$ is the predicted probability of $y$ being the $k$th-best prediction for data point $X$.

$$\arg\max_\Theta \; \mathbb{E}_{X,y\sim\mathcal{D}} \left[\log\left(\mathbb{E}_{k\sim P_K}\left[\sum_{m=1}^k \boldsymbol{P}_{m,y}\right]\right)\right] \tag{1}$$

To evaluate the probability of $y$ to be the top-1 prediction, we can simply use $\mathrm{softmax}_y(f_\Theta(X))$. However, $k > 1$ requires more consideration. Here, we require probability scores $\boldsymbol{P}_{k,c}$ for the $k$th prediction over classes $c \in \mathbb{C}$, where $\sum_{c=1}^n \boldsymbol{P}_{k,c} = 1$ (i.e., it $\boldsymbol{P}$ is row stochastic) and ideally additionally $\sum_{k=1}^n \boldsymbol{P}_{k,c} = 1$ (i.e., it $\boldsymbol{P}$ is also column stochastic and thus doubly stochastic.) With this, we can optimize our model by minimizing the following loss

$$\mathcal{L}_{\text{top}-k}(X, y) = -\log\left(\sum_{k=1}^n P_K(k) \cdot \left(\sum_{m=1}^k \boldsymbol{P}_{m,y}(f_\Theta(X))\right)\right) \tag{2}$$

which is the cross entropy over the probabilities that the true class is among the top-$k$ class for each possible $k$. Note that $\sum_{k=1}^n P_K(k) = 1$.

To compute $\boldsymbol{P}_{k,c}$, we require a function mapping from a vector of real-valued scores to an (ideally) doubly stochastic matrix $\boldsymbol{P}$. The most suitable for this are the differentiable relaxations of the sorting and ranking functions, which produce differentiable permutation matrices $\boldsymbol{P}$, which we introduced in Section 2. We build on these approximations to propose instances of top-$k$ learning losses and extend differentiable sorting networks to differentiable top-$k$ networks, as just finding the top-$k$ scores is computationally cheaper than sorting all elements and reduces the approximation error.

## 3.1 TOP-$k$ PROBABILITY MATRICES

The discussed differentiable sorting algorithms produce relaxed permutation matrices of size $n \times n$. However, for top-$k$ classification learning, we require only the top $k$ rows for the number $k$ of top-ranked classes to consider. Here, $k$ is the largest $k$ that is considered for the objective, i.e., where $P_K(k) > 0$. As $n \gg k$, producing a $k \times n$ matrix instead of a $n \times n$ matrix is much faster.

For *NeuralSort and SoftSort*, it is possible to simply compute only the top rows, as the algorithm is defined row-wise.

For the *differentiable Sinkhorn sorting algorithm*, it is not directly possible to improve the runtime, as in each Sinkhorn iteration the full matrix is required.

For *differentiable sorting networks*, it is (via a bi-directional evaluation) possible to reduce the cost from $\mathcal{O}(n^2 \log^2(n))$ to $\mathcal{O}(nk \log^2(n))$. Here, it is important to note the shape and order of multiplications for obtaining $P$. As we only need those elements, which are (after the last layer of the sorting network) at the top $k$ ranks that we want to consider, we can omit all remaining rows of the permutation matrix of the last layer (layer $t$) and thus it is only of size $(k \times n)$.

$$\underbrace{(k \times n)}_{P} = \underbrace{(k \times n)}_{\text{layer } t} \underbrace{(n \times n)}_{\text{layer } t-1} \dots \underbrace{(n \times n)}_{\text{layer } 1} \tag{3}$$

Note that during execution of the sorting network, $P$ is conventionally computed from layer 1 to layer $t$, i.e., from right to left. If we computed it in this order, we would only save a tiny fraction of the computational cost and only during the last layer. Thus, we propose to execute the differentiable sorting network, save the values that populate the (sparse) $n \times n$ layer-wise permutation matrices, and compute $P$ in a second pass from the back to the front, i.e., from layer $t$ to layer 1, or from left to right in Equation 3. This allows executing $t$ dense-sparse matrix multiplications with dense $k \times n$ matrices and sparse $n \times n$ matrices instead of dense $n \times n$ and sparse $n \times n$ matrices. With this, we reduce the asymptotic complexity from $\mathcal{O}(n^2 \log^2(n))$ to $\mathcal{O}(nk \log^2(n))$.

### 3.1.1 DIFFERENTIABLE TOP-$k$ NETWORKS

As only the top-$k$ rows of a relaxed permutation matrix are required for top-$k$ classification learning, it is possible to improve the efficiency of computing the top-$k$ probability distribution via differentiable sorting networks by reducing the number of differentiable layers and comparators. Thus, we propose differentiable top-$k$ networks, which relax selection networks in analogy to how differentiable sorting networks relax sorting networks. Selection networks are networks that select only the top-$k$ out of $n$ elements (Knuth, 1998b). We propose splitter selection networks (SSN), a novel class of selection networks that requires only $\mathcal{O}(\log n)$ layers (instead of the $\mathcal{O}(\log^2 n)$ layers for sorting networks) which makes top-$k$ supervision with differentiable top-$k$ networks more efficient and reduces the error (which is introduced in each layer.) SSNs follow the idea that the input is split into locally sorted sublists and then all wires that are not candidates to be among the global top-$k$ can be eliminated. For example, for $n = 1024, k = 5$, SSNs require only 22 layers, while the best previous selection network requires 34 layers and full sorting (with a bitonic network) requires even 55 layers. For $n = 10450, k = 5$ (i.e., for ImageNet-21K-P), SNNs require 27 layers, the best previous requires 50 layers, and full sorting requires 105 layers. In addition, the layers of SSNs are less computationally expensive than those of the bitonic sorting network. Details on SSNs, as well as their full construction, can be found in Supplementary Material B. Concluding, the contribution of differentiable top-$k$ networks is two-fold: first, we propose a novel kind of selection networks that needs fewer layers, and second, we relax those similarly to differentiable sorting networks.

## 3.2 Implementation Details

Despite those performance improvements, evaluating the differentiable ranking operators still requires a considerable amount of computational effort for large numbers of classes. Especially if the number $n$ of elements to be ranked is $n = 1\,000$ (ImageNet-1K) or even $n > 10\,000$ (ImageNet-21K-P), the differentiable ranking operators can dominate the overall computational costs. In addition, for large numbers $n$ of elements to be ranked, the performance of differentiable ranking operators decreases as differentially ranking more elements naturally introduces larger errors (Grover et al., 2019; Prillo & Eisenschlos, 2020; Cuturi et al., 2019; Petersen et al., 2021). Thus, we reduce the number of outputs to be ranked differentially by only considering those classes (for each input) that have a score among the top-$m$ scores. For this, we make sure that the ground truth class is among those top-$m$ scores, by replacing the lowest of the top-$m$ scores by the ground truth class, if necessary. For $n = 1000$, we choose $m = 16$, and for $n > 10\,000$, we choose $m = 50$. We find that this greatly improves training performance.

Because the differentiable ranking operators are (by their nature of being differentiable) only approximations to the hard ranking operator, they each have their characteristics and inconsistencies. Thus, for training models from scratch, we replace the top-1 component of the loss by the regular softmax, which has a better and more consistent behavior. This guides the other loss if the differentiable ranking operator behaves inconsistently. To avoid the top-$k$ components affecting the guiding softmax component and avoid probabilities greater than 1 in $p$, we can separate the cross-entropy into a mixture of the softmax cross-entropy (for the top-1 component) and the top-$k$ cross-entropy (for the top-$k \geq 2$ components) as follows:

$$
\mathcal{L}_{\mathrm{sm+top}-k}(X, y) = P_K(1) \cdot \mathrm{SoftmaxCrossEntropyLoss}(f_\Theta(X), y) \tag{4}
$$
$$
- (1 - P_K(1)) \cdot \log\left(\sum_{k=2}^{n} P_K(k) \cdot \left(\sum_{m=1}^{k} \boldsymbol{P}_{m,y}(f_\Theta(X))\right)\right)
$$

## 4 Related Work

We structure the related work into three broad sections: works that derive and apply differentiable top-$k$ operators, works that use ranking and top-$k$ training objectives in general, and works that present classic selection networks.

### 4.1 Differentiable Top-$k$ Operators

Grover et al. (2019) include an experiment where they use the NeuralSort differentiable top-$k$ operator for $k$NN learning. Cuturi et al. (2019), Blondel et al. (2020b), and Petersen et al. (2021) each apply their differentiable sorting and ranking methods to top-$k$ supervision with $k = 1$.

Xie et al. (2020b) propose a differentiable top-$k$ operator based on optimal transport and the Sinkhorn algorithm (Cuturi, 2013). They apply their method to $k$-nearest-neighbor learning ($k$NN), differential beam search with sorted soft top-$k$, and top-$k$ attention for machine translation. Cordonnier et al. (2021) use perturbed optimizers (Blondel et al., 2020a) to derive a differentiable top-$k$ operator, which they use for differentiable image patch selection. Lee et al. (2021) propose using NeuralSort for a differentiable top-$k$ operator to produce differentiable ranking metrics for recommender systems. Goyal et al. (2018) propose a continuous top-$k$ operator for differentiable beam search. Pietruszka et al. (2020) propose the differentiable successive halving top-$k$ operator to approximate the normalized Chamfer Cosine Similarity ($nCCS@k$).

### 4.2 Ranking and Top-$k$ Training Objectives

Fan et al. (2017) propose the "average top-$k$" loss, an aggregate loss that averages over the $k$ largest individual losses of a training data set. They apply this aggregate loss to SVMs for classification tasks. Note that this is not a differentiable top-$k$ loss in the sense of this work. Instead, the top-$k$ is not differentiable and used for deciding which data points' losses are aggregated into the loss.

Lapin et al. (2015; 2016) propose relaxed top-$k$ surrogate error functions for multiclass SVMs. Inspired by learning-to-rank losses, they propose top-$k$ calibration, a top-$k$ hinge loss, a top-$k$ entropy

loss, as well as a truncated top-$k$ entropy loss. They apply their method to multiclass SVMs and learn via stochastic dual coordinate ascent (SDCA).

Berrada et al. (2018) build on these ideas and propose smooth loss functions for deep top-$k$ classification. Their surrogate top-$k$ loss achieves good performance on the CIFAR-100 and ImageNet1K tasks. While their method does not improve performance on the raw data sets in comparison to the strong Softmax Cross-Entropy baseline, in settings of label noise and data set subsets, they improve classification accuracy. Specifically, with label noise of $20\%$ or more on CIFAR-100, they improve top-1 and top-5 accuracy and for subsets of ImageNet1K of up to $50\%$ they improve top-5 accuracy. This work is closest to ours in the sense that our goal is to improve learning of neural networks. However, in contrast to (Berrada et al., 2018), our method improves classification accuracy in unmodified settings. In our experiments, for the special case of $k$ being a concrete integer and not being drawn from a distribution, we provide comparisons to the smooth top-$k$ surrogate loss.

Yang & Koyejo (2020) provide a theoretical analysis of top-$k$ surrogate losses as well as produce a new surrogate top-$k$ loss, which they evaluate in synthetic data experiments.

A related idea is set-valued classification, where a set of labels is predicted. We refer to Chzhen et al. (2021) for an extensive overview. We note that our goal is not to predict a set of labels, but instead we return a score for each class corresponding to a ranking, where only one class can correspond to the ground truth.

### 4.3 SELECTION NETWORKS

Previous selection networks have been proposed by, i.a., (Wah & Chen, 1984; Zazon-Ivry & Codish, 2012; Karpiński & Piotrów, 2015). All of these are based on classic divide-and-conquer sorting networks, which recursively sort subsequences and merge them. In selection networks, during merging, only the top-$k$ elements are merged instead of the full (sorted) subsequences. In comparison to those earlier works, we propose a new class of selection networks, which achieve tighter bounds (for $k \ll n$), and additionally we continuously relax them.

## 5 EXPERIMENTS

### 5.1 SETUP

We evaluate the proposed top-$k$ classification loss for four differentiable ranking operators on CIFAR-100, ImageNet-1K, as well as the winter 2021 edition of ImageNet-21K-P. CIFAR-100 may be considered a small-scale data set with only 100 classes. We use CIFAR-100 to train a ResNet18 model (He et al., 2016) from scratch and show the impact of the proposed loss function on the top-1 and top-5 accuracy. In comparison, ImageNet-1K and ImageNet-21K-P provide rather large-scale data sets with $1\,000$ and $10\,450$ classes, respectively. To avoid the unreasonable carbon-footprint of training many models from scratch, we decided to exclusively use publicly available backbones for all ImageNet experiments. This has the additional benefit of allowing more settings, making our work easily reproducible, and allowing to perform multiple runs on different seeds to improve the statistical significance of the results. For ImageNet-1K, we use two publicly available state-of-the-art architectures as backbones: First, the (four) ResNeXt-101 WSL architectures by Mahajan et al. (2018), which were pretrained in a weakly-supervised fashion on a billion-scale data set from Instagram. Second, the Noisy Student EfficientNet-L2 (Xie et al., 2020a), which was pretrained on the unlabeled JFT-300M data set (Sun et al., 2017). For ResNeXt-101 WSL, we extract $2\,048$-dimensional embeddings and for the Noisy Student EfficientNet-L2, we extract $5\,504$-dimensional embeddings of ImageNet-1K and fine-tune on them.

We apply the proposed loss in combination with various available differentiable sorting and ranking approaches, namely NeuralSort, SoftSort, SinkhornSort, and DiffSortNets. The most important hyperparameter for each method is the temperature (or reparameterized as the inverse temperature). To determine the optimal temperature for each approach via grid search at a resolution of factor 2. For training, we use the Adam optimizer (Kingma & Ba, 2015). For training on CIFAR-100 from scratch, we train for up to 200 epochs with a batch size of 100 at a learning rate of $10^{-3}$. For ImageNet-1K, we train for up to 100 epochs at a batch size of 500 and a learning rate of $10^{-4.5}$. For ImageNet-21K-P, we train for up to 40 epochs at a batch size of 500 and a learning rate of $10^{-4}$.

| Method | | Public | Top-1 acc. | Top-5 acc. |
|---|---|---|---|---|
| ResNet50 | (He et al., 2015) | ✓ | 79.26 | 94.75 |
| ResNet152 | (He et al., 2015) | ✓ | 80.62 | 95.51 |
| ResNeXt-101 32x48d WSL | (Mahajan et al., 2018) | ✓ | 85.43 | 97.57 |
| ViT-L/16 | (Dosovitskiy et al., 2021) | ✓ | 87.76 | — |
| Noisy Student EfficientNet-L2 | (Xie et al., 2020a) | ✓ | **88.35** | **98.65** |
| BiT-L | (Kolesnikov et al., 2020) | ✗ | 87.54 | 98.46 |
| CLIP (w/ Noisy Student EffNet-L2) | (Radford et al., 2021) | ✗ | ≈ 88.4 | — |
| ViT-H/14 | (Dosovitskiy et al., 2021) | ✗ | 88.55 | — |
| ALIGN (EfficientNet-L2) | (Jia et al., 2021) | ✗ | 88.64 | **98.67** |
| Meta Pseudo Labels (EfficientNet-L2) | (Pham et al., 2021) | ✗ | 90.20 | ≈ 98.8 |
| ViT-G/14 | (Zhai et al., 2021) | ✗ | 90.45 | — |
| CoAtNet-7 | (Dai et al., 2021) | ✗ | **90.88** | — |
| ResNeXt-101 32x48d WSL  (used as backbone below) | | | 86.06 | 97.80 |
| Top-$k$ SinkhornSort | | | **86.29** | 97.97 |
| Top-$k$ SinkhornSort  (Top-5 focussed) | | | 86.18 | 97.99 |
| Top-$k$ SinkhornSort  (Equal focus on $k$s from 1 to 5) | | | 86.22 | 97.99 |
| Top-$k$ DiffSortNets | | | 86.24 | 97.94 |
| Top-$k$ DiffSortNets  (Top-5 focussed) | | | 86.04 | 97.98 |
| Top-$k$ DiffSortNets  (Equal focus on $k$s from 1 to 5) | | | 86.21 | **98.00** |
| Noisy Student EfficientNet-L2  (used as backbone below) | | | 88.33 | 98.65 |
| Top-$k$ SinkhornSort | | | 88.32 | 98.66 |
| Top-$k$ DiffSortNets | | | **88.37** | **98.68** |

Table 1: ImageNet-1K result comparison to state-of-the-art. Among the overall best performing differentiable sorting / ranking methods, almost all results in reasonable settings outperform their respective baseline on Top-1 and Top-5 accuracy. For publicly available models / backbones, we achieve a new state-of-the-art for top-1 and top-5 accuracy. Our results are averaged over 10 runs.

We use early stopping and found that these settings lead to convergence in all settings. As baselines, we use the respective original models, softmax cross-entropy, as well as learning with the smooth surrogate top-$k$ loss (Berrada et al., 2018).

## 5.2 COMPARISON TO THE STATE-OF-THE-ART

We compare the proposed results to current state-of-the-art methods in Table 1. We focus on methods that are publicly available and build upon two of the best performing models, namely Noisy Student EfficientNet-L2 (Xie et al., 2020a), and ResNeXt-101 32x48d WSL (Mahajan et al., 2018). Using both backbones, we achieve improvements on both metrics, and when fine-tuning on the Noisy Student EfficientNet-L2, we achieve a new state-of-the-art for publicly available models.

In Figure 2, we demonstrate our improvements on the four model sizes of ResNeXt-101 WSL (32x8d, 32x16d, 32x32d, 32x48d). Our method improves the model in all settings.

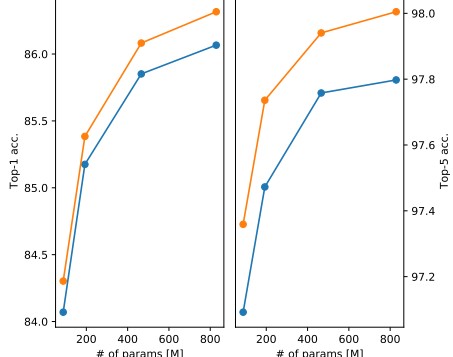

Figure 2: ImageNet-1K accuracy improvements for all ResNeXt-101 WSL model sizes (32x8d, 32x16d, 32x32d, 32x48d). Blue is the original model and orange is with top-$k$ fine-tuning.

**Significance Tests.** To evaluate the significance of the results, we perform a t-test (with significance level of 0.01). We find that our model is significantly better than the original model on both top-1 and top-5 accuracy metrics. Comparing to the observed accuracies of the baseline (88.33 | 98.65), DiffSortNets are significantly better (p=0.00001 | 0.00005). Comparing to the reported accuracies of the baseline (88.35 | 98.65), DiffSortNets are also significantly better (p=0.00087 | 0.00005).

## 5.3 IMPACT OF THE DISTRIBUTION $P_K$ AND DIFFERENTIABLE SORTING METHODS

We start by demonstrating the impact of $P_K$, which is the distribution from which we draw $k$. Let us first consider the case where $k$ is 5 with probability $\alpha$ and 1 with probability $1 - \alpha$, i.e., $P_K =$

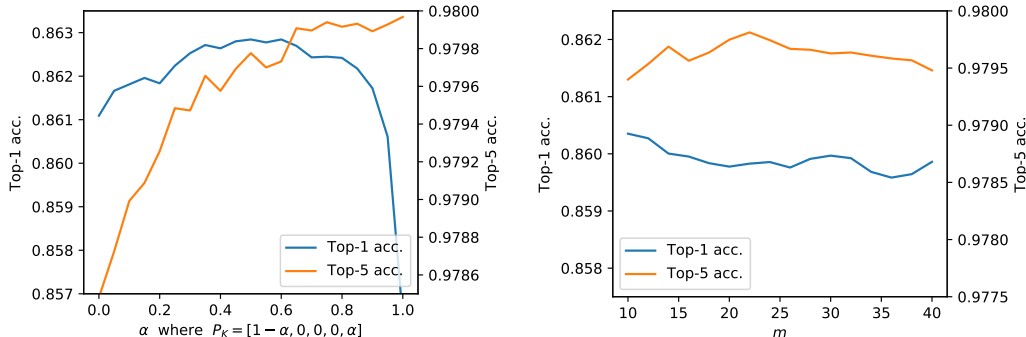

Figure 3: Effects of varying the ratio between top-1 and top-5 (left) and varying the size of differentially ranked subset $m$. Both experiments are done with the differentiable Sinkhorn ranking algorithm (Cuturi et al., 2019). On the left, $m = 16$, on the right, $\alpha = 0.75$. Averaged over 5 runs.

$[1 - \alpha, 0, 0, 0, \alpha]$. In Figure 3 (left), we demonstrate the impact that changing $\alpha$, i.e., transitioning from a pure top-1 loss to a pure top-5 loss, has on fine-tuning ResNeXt-101 WSL with our loss using the SinkhornSort algorithm. Increasing the weight of the top-5 component does not only increase the top-5 accuracy but also improves the top-1 accuracy up to around $60\%$ top-5; when using only $k = 5$, the top-1 accuracy drastically decays as the incentive for the true class to be at the top-1 position vanishes (or is only indirectly given by being among the top-5.) While the top-5 accuracy in this plot is largest for a pure top-5 loss, this generally only applies to the Sinkhorn algorithm and overall training is stable if a pure top-5 is avoided. This can also be seen on the left of Table 2.

In Table 2, we consider more additional settings with all diff. ranking methods. Specifically, we compare four notable settings: $[.5, 0, 0, 0, .5]$, i.e., equally weighted top-1 and top-5; $[.25, 0, 0, 0, .75]$ and $[.1, 0, 0, 0, .9]$, i.e., top-5 has larger weights; $[.2, .2, .2, .2, .2]$, i.e., the case of having an equal weight of $0.2$ for top-1 to top-5. The $[.5, 0, 0, 0, .5]$ setting is a rather canonical setting which usually performs well on both metrics, while the others tend to favor top-5. In the $[.5, 0, 0, 0, .5]$ setting, all sorting methods improve upon the softmax baseline on both top-1 and top-5 accuracy. When increasing the weight of the top-5 component, the top-5 generally improves while top-1 decays.

Here we find the main insight of this paper: the best performance cannot be achieved by optimizing top-$k$ for only a single $k$, but instead relaxing this constraint improves performance for all metrics.

| ImageNet-1K / $P_K$ | $[1,0,0,0,0]$ | $[0,0,0,0,1]$ | $[.5,0,0,0,.5]$ | $[.25,0,0,0,.75]$ | $[.1,0,0,0,.9]$ | $[.2,.2,.2,.2,.2]$ |
|---|---|---|---|---|---|---|
| Softmax (baseline) | 86.06 \| 97.795 | — | — | — | — | — |
| Smooth top-$k$ loss (Berrada et al., 2018) | 85.15 \| 97.540 | — | — | — | — | — |
| NeuralSort | — | 33.37 \| 94.748 | 86.30 \| 97.896 | 34.26 \| 95.410 | 34.32 \| 94.889 | 85.75 \| 97.865 |
| SoftSort | — | 18.23 \| 94.965 | 86.26 \| 97.963 | 86.16 \| 97.954 | 27.30 \| 95.915 | 86.18 \| 97.979 |
| SinkhornSort | — | 85.65 \| 97.991 | **86.29** \| 97.971 | 86.24 \| 97.989 | 86.18 \| 97.987 | 86.22 \| 97.989 |
| DiffSortNets | — | 69.05 \| 97.389 | 86.24 \| 97.937 | 86.15 \| 97.936 | 86.04 \| 97.980 | 86.21 \| **98.003** |

Table 2: ImageNet-1K results for fine-tuning the head of ResNeXt-101 32x48d WSL (Mahajan et al., 2018) averaged over 10 runs. The displayed metrics are Top-1 \| Top-5 accuracies.

| ImageNet-21K-P / $P_K$ | $[1,0,0,0,0]$ | $[0,0,0,0,1]$ | $[.5,0,0,0,.5]$ | $[.25,0,0,0,.75]$ | $[.1,0,0,0,.9]$ | $[.2,.2,.2,.2,.2]$ |
|---|---|---|---|---|---|---|
| Softmax (baseline) | 39.29 \| 69.63 | — | — | — | — | — |
| Smooth top-$k$ loss (Berrada et al., 2018) | 34.03 \| 65.56 | — | — | — | — | — |
| NeuralSort | — | 15.87 \| 33.81 | 37.85 \| 68.08 | 36.16 \| 67.60 | 33.02 \| 67.29 | 37.09 \| 67.90 |
| SoftSort | — | 33.61 \| 69.82 | 39.93 \| 70.63 | 39.08 \| 70.27 | 37.78 \| 70.07 | 39.68 \| 70.57 |
| SinkhornSort | — | 36.93 \| 69.80 | 39.85 \| 70.56 | 39.21 \| 70.41 | 38.42 \| 70.12 | 39.22 \| 70.49 |
| DiffSortNets | — | 35.96 \| 69.76 | **40.22 \| 70.88** | 39.56 \| 70.58 | 38.48 \| 70.25 | 39.69 \| 70.69 |

Table 3: ImageNet-21K-P results for fine-tuning the head of ResNeXt-101 32x48d WSL (Mahajan et al., 2018). The metrics are Top-1 \| Top-5 accuracy averaged over 2 seeds.

| CIFAR-100 / $P_K$ | $[1,0,0,0,0]$ | $[0,0,0,0,1]$ | $[.5,0,0,0,.5]$ | $[.25,0,0,0,.75]$ | $[.1,0,0,0,.9]$ | $[.2,.2,.2,.2,.2]$ |
|---|---|---|---|---|---|---|
| Softmax (baseline) | 61.27 \| 85.31 | — | — | — | — | — |
| Smooth top-$k$ loss (Berrada et al., 2018) | 53.07 \| 85.23 | — | — | — | — | — |
| NeuralSort | — | 22.58 \| 84.41 | 61.12 \| 86.47 | 61.07 \| 87.23 | 52.57 \| 85.76 | 61.46 \| 86.03 |
| SoftSort | — | 1.01 \| 5.09 | 61.17 \| 83.95 | 61.05 \| 83.10 | 58.16 \| 79.26 | 61.53 \| 82.39 |
| SinkhornSort | — | 55.62 \| 87.04 | 61.34 \| 86.38 | 61.50 \| 86.68 | 57.35 \| 86.34 | 61.89 \| 86.94 |
| DiffSortNets | — | 52.81 \| 84.21 | 60.07 \| 86.44 | 61.57 \| 86.51 | 61.74 \| **87.22** | **62.00** \| 86.73 |

Table 4: CIFAR-100 results for training a ResNet18 from scratch averaged over 2 seeds.

Comparing the differentiable ranking methods, we can find the overall trend that SoftSort outperforms NeuralSort, and that SinkhornSort as well as DiffSortNets perform best. We can see that some sorting algorithms are more sensitive to the overall $P_K$ than others: Whereas SinkhornSort (Cuturi et al., 2019) and DiffSortNets (Petersen et al., 2021) continuously outperform the softmax baseline, NeuralSort (Grover et al., 2019) and SoftSort (Prillo & Eisenschlos, 2020) tend to collapse when over-weighting the top-5 components.

Comparing the performance on the medium-scale ImageNet-1K to the larger ImageNet-21K-P in Table 3 we observe a similar pattern. Here, again, using the top-$k$ loss alone is not enough to significantly increase accuracy, but combining both, top-1 and top-$k$ loss helps to improve accuracy on both reported metrics. While NeuralSort struggles with a good loss for this large-scale ranking problem and stays below the softmax baseline, DiffSortNets (Petersen et al., 2021) provide the best top-1 and top-5 accuracy with 40.22% and 70.88%, respectively.

We note that we do not claim that all settings (especially all differentiable sorting methods) improve the classification performance. Instead, we include all methods and also additional settings to demonstrate the capabilities and limitations of each differentiable sorting method.

It is notable that SinkhornSort overall achieves the most robust training behavior, while also being by far the slowest sorting method and thus potentially slowing down training drastically, especially when the task is only fine-tuning. SinkhornSort tends to require more Sinkhorn iterations towards the end of training. DiffSortNets are considerably faster, especially it is possible to only compute the top-$k$ probability matrices and because of our advances for more efficient selection networks.

## 5.4 DIFFERENTIABLE RANKING SET SIZE $m$

We consider how accuracy is affected by varying the number of scores $m$ to be differentially ranked. Generally, the runtime of differentiable top-$k$ operators depends between linearly and cubic on $m$; thus it is important to choose an adequate value for $m$. The choice of $m$ between 10 and 40 has only a moderate impact on the accuracy as can be seen in Figure 3 (right). However, when setting $m$ to large values such as 1 000 or larger, we observe that the differentiable sorting methods tend to become unstable. We note that we did not specifically tune $m$, and that better performance can be achieved by fine-tuning $m$, as displayed in the plot.

## 5.5 TRAINING FROM SCRATCH

Finally, we demonstrate that the proposed loss can also be used to train a network from scratch. As reference baseline, we train a ResNet18 from scratch on CIFAR-100, considering the same settings as above. The results are shown in Table 4. Again, we find that training with top-$k$ alone slightly improves the top-5 but not the top-1 accuracy, whereas a weighted combination of both losses can significantly improve the performance on both metrics. Notably, here, $[.2,.2,.2,.2,.2]$ with DiffSortNets yields the best results on top-1 accuracy.

In Supplementary Material A, an extension to learning with top-10 and top-20 components on ImageNet-21K-P can be found.

## 6 CONCLUSION

In this work, we presented a novel loss, which relaxes the assumption of using a fixed $k$ for top-$k$ classification learning. For this, we leveraged recent differentiable sorting and ranking operators. We performed an array of experiments to explore different top-$k$ classification learning settings and achieved a state-of-the-art on ImageNet for publicly available models.

REPRODUCIBILITY STATEMENT

The source code for the experiments will be made available upon publication. We only use publicly available resources, and each experiment can be reproduced using small-scale hardware. We used (wherever possible) multiple seeds for our experiments.

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

## A  EXTENSION TO TOP-10 AND TOP-20

We further extend the training settings, measuring the impact of top-10 and top-20 components on the large-scale ImageNet-21K-P dataset. The results are diplayed in Table 5, where we report top-1, top-5, top-10, and top-20 accuracy for all configurations. Again, we observe that $50\%$ top-1 and $50\%$ top-$k$ produces the overall best performance and that training with top-5 yields the best top-1, top-5, and top-10 accuracy. We observe that the performance decays for top-20 components because (even among $10\,450$ classes) there are virtually no top-20 ambiguities, and artifacts of differentiable sorting methods can cause adverse effects. Note that top-10 ambiguities do exist in ImageNet-21K-P, e.g., there are 11 class hierarchy levels Ridnik et al. (2021).

| *IN-21K-P* / $P_K$ (@5) | $[1, 0, ...]$ | $[.5, 0, ..., 0, .5]$ | $[.25, 0, ..., 0, .75]$ | $[.1, 0, ..., 0, .9]$ |
|---|---|---|---|---|
| | len=5 | len=5 | len=5 | len=5 |
| Softmax (baseline) | 39.29 \| 69.63 \| 78.55 \| 85.33 | — | — | — |
| NeuralSort | — | 37.85 \| 68.08 \| 77.22 \| 84.21 | 36.16 \| 67.60 \| 76.96 \| 84.08 | 33.02 \| 67.29 \| 76.88 \| 84.05 |
| SoftSort | — | 39.93 \| 70.63 \| 79.45 \| 85.96 | 39.08 \| 70.27 \| 79.29 \| 85.94 | 37.78 \| 70.07 \| 79.19 \| 85.87 |
| SinkhornSort | — | 39.85 \| 70.56 \| 79.53 \| 86.13 | 39.21 \| 70.41 \| 79.54 \| 86.18 | 38.42 \| 70.12 \| 79.44 \| 86.12 |
| DiffSortNets | — | **40.22** \| **70.88** \| **79.54** \| 86.03 | 39.56 \| 70.58 \| 79.44 \| 86.01 | 38.48 \| 70.25 \| 79.29 \| 85.90 |
| *IN-21K-P* / $P_K$ (@10) | $[1, 0, ...]$ | $[.5, 0, ..., 0, .5]$ | $[.25, 0, ..., 0, .75]$ | $[.1, 0, ..., 0, .9]$ |
| | len=10 | len=10 | len=10 | len=10 |
| Softmax (baseline) | 39.33 \| 69.62 \| 78.55 \| 85.36 | — | — | — |
| NeuralSort | — | 37.22 \| 67.02 \| 76.75 \| 84.10 | 34.59 \| 66.09 \| 76.46 \| 84.01 | 29.60 \| 65.16 \| 76.26 \| 84.01 |
| SoftSort | — | 39.26 \| 69.52 \| 79.13 \| 85.93 | 37.71 \| 68.56 \| 78.71 \| 85.78 | 33.68 \| 67.35 \| 78.43 \| 85.70 |
| SinkhornSort | — | 39.65 \| 70.25 \| 79.47 \| 86.22 | 38.90 \| 69.91 \| 79.41 \| **86.25** | 37.98 \| 69.57 \| 79.33 \| 86.16 |
| DiffSortNets | — | 39.92 \| 70.13 \| 79.38 \| 86.02 | 39.10 \| 69.60 \| 79.21 \| 86.03 | 37.88 \| 69.07 \| 79.04 \| 85.91 |
| *IN-21K-P* / $P_K$ (@20) | $[1, 0, ...]$ | $[.5, 0, ..., 0, .5]$ | $[.25, 0, ..., 0, .75]$ | $[.1, 0, ..., 0, .9]$ |
| | len=20 | len=20 | len=20 | len=20 |
| Softmax (baseline) | 39.33 \| 69.62 \| 78.55 \| 85.36 | — | — | — |
| NeuralSort | — | 36.32 \| 65.33 \| 75.82 \| 83.99 | 33.00 \| 62.99 \| 74.84 \| 83.83 | 27.35 \| 60.34 \| 74.02 \| 83.77 |
| SoftSort | — | 38.04 \| 65.98 \| 77.17 \| 85.45 | 34.30 \| 62.89 \| 76.03 \| 85.19 | 24.02 \| 56.35 \| 74.32 \| 84.82 |
| SinkhornSort | — | 39.76 \| 69.76 \| 79.17 \| 86.17 | 38.77 \| 69.18 \| 78.99 \| 86.20 | 37.71 \| 68.68 \| 78.86 \| 86.16 |
| DiffSortNets | — | 39.54 \| 68.59 \| 77.95 \| 85.49 | 38.62 \| 67.62 \| 77.43 \| 85.37 | 37.46 \| 66.80 \| 77.01 \| 85.17 |

Table 5: ImageNet 21K with top-5, top-10 and top-20 components. The displayed metrics per column are (Top-1 | Top-5 | Top-10 | Top-20).

## B  SPLITTER SELECTION NETWORKS

Similar to a sorting network, a *selection network* is generally a comparator network and hence it consists of wires (or lanes) carrying values and comparators (or conditional swap devices) connecting pairs of wires. A comparator swaps the values on the wires it connects if they are not in a desired order. However, in contrast to a sorting network, which sorts all the values carried by its wires, a $(k, n)$ selection network, which has $n$ wires, moves the $k \leq n$ largest (or, alternatively, the $k$ smallest) values to a specific set of wires (Knuth, 1998b), most conveniently consecutive wires on one side of the wire array. Note that the notion of a selection network usually does not require that the selected values are sorted. However, in our context it is preferable that they are, so that $P_K$ can easily be applied, and the selection networks discussed below all have this property.

Clearly, any sorting network could be used as a selection network, namely by focusing only on the top $k$ (or bottom $k$) wires. However, especially if $k$ is small compared to $n$, it is possible to construct selection networks with smaller size (i.e. fewer comparators) and often lower depth (i.e. a smaller number of layers, where a layer is a set of comparators that can be executed in parallel).

A core idea of constructing selection networks was proposed in (Wah & Chen, 1984), based on the odd-even merge and bitonic sorting networks (Batcher, 1968): partition the $n$ wires into subsets of at least $k$ wires (preferably $2^{\lceil \log_2(k) \rceil}$ wires per subset) and sort each subset with odd-even mergesort. Then merge the (sorted) top $k$ elements of each subsets with bitonic merge, thus halving the number of (sorted) subsets. Repeat merging pairs of (sorted) subsets until only a single (sorted) subset remains, the top $k$ elements of which are the desired selection. This approach requires $\frac{1}{2}\lceil \log_2(k) \rceil (\lceil \log_2(k) \rceil + 1) + (\lceil \log_2(n) \rceil - \lceil \log_2(k) \rceil)(\lceil \log_2(k) \rceil + 1)$ layers.



Figure 4: Minimum ranks after a splitter cascade resulting from the transitive closure of the swaps.

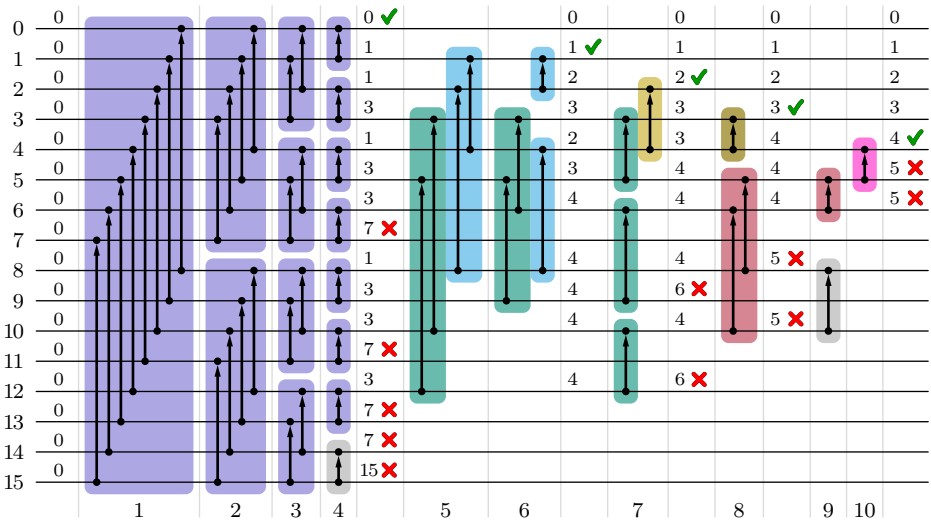

Figure 5: A $(5, 16)$ selection network constructed with the method described in the text. The numbers on the wires are the minimum ranks (starting at 0) that can be occupied by the values on these wires. Red crosses mark where wires can be excluded, green check marks where a top rank is determined. Swaps in blocks of equal color belong to the same splitter cascade. Swaps in gray boxes would be needed for full splitter cascades, but are not needed to determine the top 5 ranks.

Improvements to this basic scheme were developed in (Zazon-Ivry & Codish, 2012; Karpiński & Piotrów, 2015) and either rely entirely on odd-even merge (Batcher, 1968) or entirely on pairwise sorting networks (Parberry, 1992). Especially selection networks based on pairwise sorting networks have advantages in terms of the size of the resulting network (i.e. number of needed comparators). However, these improvements do not change the depth of the networks, that is, the number of layers, which is most important in the context considered here.

Our own selection network construction draws on this work by focussing on a specific ingredient of pairwise sorting networks, namely a so-called splitter (which happens to be identical to a single bitonic merge layer, but for our purposes it is more comprehensible to refer to it as a splitter). A splitter for a list of $m$ wires having indices $[\ell_0, \ldots, \ell_{m-1}]$ has comparators connecting wires $\ell_i$ and $\ell_{i+s}$ where $s = \lceil \log_2(m) \rceil - 1$ for $i \in \{0, \ldots, m - 1 - 2^{\lceil \log_2(m) \rceil) - 1}\}$.

A pairwise sorting network starts with what we call a *splitter cascade*. That is, an initial splitter partitions the input wires into subsets of (roughly) equal size. Each subset is split recursively until wire singletons result (Zazon-Ivry & Codish, 2012). An example of such a splitter cascade is shown in Figure 4 for 8 wires and in purple color for 16 wires in Figure 5 (arrows point to where the larger value is desired).

After a splitter cascade, the value carried by wire $\ell_i$ has a minimum rank of $r = 2^{b(i)} - 1$, where $b(i)$ counts the number of set bits in the binary number representation of $i$. This minimum rank results from the transitivity of the swap operations in the splitter cascade, as is illustrated in Figure 4 for 8 wires: By following upward paths (in splitters to the left) through the splitter cascade, one can find

| $k$ | full | odd-even/pairwise/bitonic selection | | | | | | | | splitter selection | | | | | | | |
|---|---|---|---|---|---|---|---|---|---|---|---|---|---|---|---|---|---|
| $n$ | sort | 1 | 2 | 3 | 4 | 5 | 6 | 7 | 8 | 1 | 2 | 3 | 4 | 5 | 6 | 7 | 8 |
| 16 | 10 | 4 | 7 | 9 | 9 | 10 | 10 | 10 | 10 | 4 | 6 | 7 | 8 | 10 | 11 | 12 | 13 |
| 1024 | 55 | 10 | 19 | 27 | 27 | 34 | 34 | 34 | 34 | 10 | 14 | 16 | 18 | 22 | 25 | 27 | 29 |
| 10450 | 105 | 14 | 27 | 39 | 39 | 50 | 50 | 50 | 50 | 14 | 18 | 20 | 23 | 27 | 30 | 32 | 34 |
| 65536 | 136 | 16 | 31 | 45 | 45 | 58 | 58 | 58 | 58 | 16 | 20 | 22 | 25 | 29 | 32 | 34 | 36 |

Table 6: Depths of sorting networks and selection networks (which are equal for odd-even, pairwise, or bitonic networks) compared to selection networks constructed with our splitter-based approach. Note that for small $n$ and comparatively large $k$ an odd-even/pairwise/bitonic selection network or even a full sorting network may be preferable (e.g. $n = 16$ and $k > 5$), but that for larger $n$ considerable savings can be obtained for small $k$, even compared to other selection networks.

for each wire $\ell_i$ exactly $r = 2^{b(i)}-1$ wires with smaller indices that must carry values no less than the value carried by wire $\ell_i$. This yields the minimum ranks shown in Figure 4 on the right.

The core idea of our selection network construction is to use splitter cascades to increase the minimum ranks of (the values carried by) wires. If such a minimum rank exceeds $k$ (or equals $k$, since we work with zero-based ranks and hence are interested in ranks $\{0, \ldots, k-1\}$), a wire can be discarded, since its value is certainly not among the top $k$. On the other hand, if there is only one wire with minimum rank 0, the top 1 value has been determined. More generally, if all minimum ranks no greater than some value $r$ occur for one wire only, the top $r + 1$ values have been determined.

We exploit this as follows: Initially all wires are assigned a minimum rank of 0, since at the beginning we do not know anything about the values they carry. We then repeat the following construction: traversing the values $r = k-1, \ldots, 0$ descendingly, we collect for each $r$ all wires with minimum rank $r$ and apply a splitter cascade to them (provided there are at least two such wires). Suppose the wires collected for a minimum rank $r$ have indices $[\ell_0, \ldots, \ell_{m(r)-1}]$. After the splitter cascade we can update the minimum rank of wire $\ell_i$ to $r + 2^{b(i)}-1$, because before the splitter cascade there is no known relationship between wires with the same minimum rank, while the splitter cascade establishes relationships between them, increasing their ranks by $2^{b(i)}-1$. The procedure of traversing the minimum ranks $k-1, \ldots, 0$ descendingly, collecting wires with the same minimum rank and applying splitter cascades to them is repeated until all minimum ranks $0, \ldots, k-1$ occur only once.

As an example, Figure 5 shows a $(5, 16)$ selection network constructed is this manner, in which the minimum ranks of the wires are indicated after certain layers as well as when certain wires can be discarded (red crosses) and when certain top ranks are determined (green check marks). Comparators belonging to the same splitter cascade are shown in the same color.

While selection networks resulting from adaptations of sorting networks (see above) have the advantage that they guarantee that their number of layers is never greater than that of a full sorting network, our approach may produce networks with more layers. However, if $k$ is sufficiently small compared to $n$ (in particular, if $k \leq \log_2(n)$), our approach can produce selection networks with considerably fewer layers, as is demonstrated in Table 6. Since in the context we consider here we can expect $k \leq \log_2(n)$, splitter-based selection networks are often superior.

