# OpenReview forum: "Differentiable Top-k Classification Learning"
_ICLR.cc/2022/Conference — ICLR 2022 Submitted_

### Official Review · Reviewer_FWmd · 2021-10-26

**Correctness:** 3
**Technical Novelty And Significance:** 2
**Empirical Novelty And Significance:** 2
**Recommendation:** 5
**Confidence:** 5

**Main Review:**

Strengths
- Set-valued classification is an important topic to cope with class ambiguity. Few works (only one as far as I know [1]) have proposed a top-k loss for neural networks and there is room for improvements
- The proposed approach is different from [1] as it relies on sorting networks to determine the set of the most likely classes rather than a purely top-k objective

Weaknesses
- A first weakness is that the contribution is quite incremental and not well justified from a theoretical point of view. Using sorting networks for top-k is an acceptable strategy from a practical point of view but a bit over-kill and not very new from a theoretical point of view. The proposal to use several values of K is also not really justified. The principle of top-K is to predict sets of fixed size contrary to other set-valued classification approaches that attempt to solve other objectives (e.g. adaptive set sizes but equal to K on average). We let the authors refer to [2] for a clear overview of the different objectives. Here, the objective is not really clear. If K is supposed to be a random variable (e.g. 50% chance of being 1 and  50% chance of being 5), that means that for the same image x, the classifier is supposed to return randomly either one class or 5 classes without any consideration with regard to the image content itself.
- another main weakness is that no significant improvement of the proposed loss over cross-entropy is shown. The reported top-K accuracy gains are not systematic and so low that they may be not statistically significant. As a first step towards a better understanding of the results, the authors should first compute some significance tests (e.g. p-values on several runs and a clear cross-validation procedure for model selection among epochs). But even so, it won’t resolve the fact that the performance gain is observed only for some specific configurations (e.g. a specific sorting network and specific values of K probabilities) and remains very low even in such advantageous conditions.

[1] Berrada, L., Zisserman, A., & Kumar, M. P. (2018). Smooth loss functions for deep top-k classification. arXiv preprint arXiv:1802.07595.
[2] Chzhen, E., Denis, C., Hebiri, M., & Lorieul, T. (2021). Set-valued classification--overview via a unified framework. arXiv preprint arXiv:2102.12318.


**Summary Of The Paper:**

The paper proposes a differentiable loss for top-K classification based on differentiable sorting networks, i.e. sorting neural networks in which basic min/max operations are replaced by smoothed versions (i.e. softmax/softmin). The main principle is to use the sorting network to estimate the probability of the rank of each class and then filter only the top-k. An extension consists in considering that k can take several possible values at random (e.g. 50% chance of being 1 and  50% chance of being 5).

The resulting loss is experimented on three datasets (CIFAR-100, ImageNet-1K and ImageNet-21K-P) and with three existing sorting networks. Performances are mainly compared to cross-entropy showing low improvements.

**Summary Of The Review:**

An interesting attempt to improve the top-K classification but consistent limitations:
(I) an incremental contribution and no clear justification of considering k as a random variable
(ii) no significant improvement of the proposed loss over cross-entropy

---

> ### Author Response · Authors · 2021-11-10
> **Response to Reviewer FWmd (1/2)**
>
> Thank you very much for your review.
> Please let us know whether you have further suggestions and questions.
>
> Thank you for pointing out the reference “Set-valued classification—overview via a unified framework” by Chzhen, E., Denis, C., Hebiri, M., & Lorieul, T. (2021), which we will include in our related work discussion.
>
> We would like to emphasize that the main principle is not “to use the sorting network to estimate the probability of the rank of each class and then filter only the top-k”.
> Instead, our main principle is to use *any* differentiable sorting / ranking operator (such as a differentiable sorting network or Sinkhorn among others) **and** relaxing $k$ to be drawn from a distribution.
> Relaxing $k$ is not an extension but instead an integral part of our method, only a special case of which is using a constant $k$.
> Using a constant $k$ already exists in the literature (e.g., Berrada et al. for a surrogate loss).
> We are unsure what “with three existing sorting networks” refers to.
> Could you please clarify this?
>
> There might have been some confusion about NeuralSort and sorting networks: We would like to clarify that NeuralSort is not based on sorting networks (contrary to the title of the NeuralSort paper) and that sorting networks are not synonymous to differentiable sorting. Differentiable sorting networks are also only one of three categories of differentiable sorting methods.
>
> We have explicitly clarified this in the revision.
>
> **We let the authors refer to [2] for a clear overview of the different objectives. Here, the objective is not really clear. If K is supposed to be a random variable (e.g. 50% chance of being 1 and 50% chance of being 5), that means that for the same image x, the classifier is supposed to return randomly either one class or 5 classes without any consideration with regard to the image content itself.**:
> We would like to point out that what we do is not set-valued classification.
> Instead, we have a top-k classification setting, which we optimize for a distribution of k.
> Framing our method as set-valued classification is problematic, as you pointed out.
> The classifier itself never decides whether it returns 1 or 5 classes, and it also does not return randomly either one class or 5 classes.
> Our model always returns a score for each class, which corresponds to a ranking, and it is up to the practitioner to decide what to do with the scores or the ranking that underlies the scores.
> For example, with $[0.5, 0, 0, 0, 0.5]$ we model, a setting in which $50\%$ of a model's users want to have a good top-1 accuracy and $50\%$ of a model's users want to have a good top-5 accuracy.
> However, this might also be the case if a practitioner wants a good top-1 model, but if it is not good at a certain data point, he/she is still happy if the correct class is among the top-5 predicted classes.
> A way to think about this is diversification: for example, an investor might want to invest $50\%$ into the stock with the best score and equally distribute the remaining $50\%$ onto the top-5 stocks.
> Given this policy, the model should be trained to maximize the expectation value of the return of the investor.
> Another case is where we want to use the top-1 prediction, but if it turns out to be incorrect, we fall back to using the next four options, which we wish to also be meaningful.
> A main finding of our work is that using such a training regime, we can also improve the conventional top-1 accuracy of our model, which is independent of how the model actually ends up being used.
>
> We added [2] to the related work section.
>
> $\to$ continued in next comment

---

> > ### Author Response · Authors · 2021-11-10
> > **Response to Reviewer FWmd (2/2)**
> >
> > $\uparrow$ continued from above.
> >
> > **another main weakness is that no significant improvement of the proposed loss over cross-entropy is shown. The reported top-K accuracy gains are not systematic and so low that they may be not statistically significant. As a first step towards a better understanding of the results, the authors should first compute some significance tests (e.g. p-values on several runs and a clear cross-validation procedure for model selection among epochs).**:
> > We agree that this is correct for some settings of NeuralSort and SoftSort, which is also expected as they also do not perform well in other differentiable sorting/ranking tasks.
> > However, for Sinkhorn Sort and DiffSortNets, we have a quite consistent improvement.
> > The fact that a top-5 loss does not yield good results is expected, and the main point of this paper: Relaxing $k$ to a distribution improves performance and makes it more stable. Everything on the left of the dividing line in Tables 2, 3, and 4 are baselines.
> > The values that are representative for our proposed method are in the rows "SinkhornSort" and "DiffSortNets" and in the columns $[.5,0,0,0,.5]$ and $[.2,.2,.2,.2,.2]$ as well as (when actively putting a larger emphasis on top-5) $[.25,0,0,0,.75], [.1,0,0,0,.9]$.
> > Especially for the top-5 accuracy for ResNeXt-101 WSL (32x48d) we reduce the error from $2.2\%$ to $2.0\%$, which is a substantial improvement (relative $10\%$).
> > Also, doubling the model size (from 32x16d to 32x32d) is around as effective as using our method (see Figure 2).
> >
> > We added a paragraph to Section 5.2 to include significance tests.
> > We find that top-$k$ learning with DiffSortNets performs significantly better and report the p-values as suggested.
> >
> > **moreover, the proposed loss is not compared to the only top-k loss existing in the literature for deep networks, i.e. the one of Berrada et al. [1]. The code is freely available and easy ti integrate in standard deep learning frameworks so that there is no reason not to test it. And to be completely fair, it should be tested by using several K simultaneously as for the proposed method…**:
> > We would like to point out that we **do** compare to the smooth top-k loss by Berrada et al. in Tables 2, 3, and 4.
> > This is the "Smooth top-k loss" method in the second row.
> > We apologize that it was potentially ambiguous and fixed this by adding a reference to Berrada et al. at the respective locations.
> > We agree that "tested by using several K simultaneously as for the proposed method" is an interesting idea for future work; however, we note that this is not immediately supported by the loss formulation, as simply adding a top-1 and a top-5 loss is not equivalent to using several k simultaneously.
> > Nevertheless, it would be an interesting future development to combine the ideas of our work (using different k) and combining it in some way with the surrogate smooth top-k loss.
> >
> > **finally, the paper is not really well written. The authors should try to pay more attention to the used machine learning terminology and also improve the English language.**
> > We are always happy about feedback and would really appreciate if you could point us to the specific terminological errors and English errors that you found such that we can fix them.
> >
> > We are looking forward to hearing back from you.

---

### Official Review · Reviewer_RKLW · 2021-11-02

**Correctness:** 3
**Technical Novelty And Significance:** 3
**Empirical Novelty And Significance:** 3
**Recommendation:** 6
**Confidence:** 4

**Main Review:**

Strengths:
- The efforts on optimizing the top-k classification learning through differentiable sorting appear novel to me. The discussion on differential sorting is comprehensive. The paper specifically discusses each of the options and how it is optimized for the studied scenario.
- Experiments are thorough
- The work is also interesting because the performance gains come only because of fine-tuning.


Weaknesses

- The second term and eqn 2 would be constant with k=5 (if only the top five rows of the P matrix are constructed). Expanding eqn2 for the given example in Fig1, the loss would be: -log(0.5 *.03 + 0.5 (0.3+0.6)), assuming Panda is the ground truth class. Consider a case of P_K [0.5 0 0 0 0.5], the equation would be -log (0.5 top1 + 0.5 (top1+top2+top3+top4+top5)). If only five columns are reconstructed and if they are column stochastic, then the sum of top1 to top5 would always be 1. Then the second term will always give a constant value. Requesting the authors to clarify this aspect.

- At first, it appears that the distribution would be a sample. However, fixed distribution is used for a set of experiments. For example, it is either [0.5 0 0 0 0.5] or [0.2 0.2 0.2 0.2 0.2] for the entire experiment. Hence, presenting it as "sampled" is confusing. The best results come when you have the top1 and the sum of the top five values. Hence, the initial discussion and intuition can be improved a bit.

- The improvements on Noisy Student EfficientNet-L2 are negligible. 88.35 to 88.36 is certainly not statistically significant. Were experiments for Table1 were also ran 10 times (like table 2?).

- Please mention the number of rows that were reconstructed for each experiment. The number of columns (m) is mentioned in the experiments but not the number of rows.

- Berrada et al. was used to train the model from scratch. It would be worth comparing their loss for fine-tuning purposes as well. I think that would be a fairer comparison.



**Summary Of The Paper:**

The paper proposes a method to employ the benefits of differential sorting methods towards top-k classification learning. The presents several experiments with sampling weights for different ranks and presents the results. The loss is used for fine-tuning in most experiments (apart from the CIFAR100 case). The method seems to give minor improvements on the ResNeXt-101 32*48d baseline.

**Summary Of The Review:**

 Although the paper brings several novel perspectives, there remain several ambiguities as well. Some additional experiments, clarifications can also strengthen the draft. Overall, in the current form, the paper is a borderline one and the final decision will depend a lot on the discussion during the rebuttal phase.

---

> ### Author Response · Authors · 2021-11-10
> **Response to Reviewer RKLW**
>
> Thank you very much for your encouraging review. We appreciate extensive comments.
> Please let us know whether you have further suggestions and questions.
>
> **The second term and eqn 2 would be constant with k=5 […]**:
> If there are only 5 classes (i.e., 5 columns in the matrix as in the visual example in Figure 1), the top-5 component is indeed constant. However this is only the case for a top-k component with k classes. In the experiments we do not use a $5\times 5$ matrix P, but instead a much larger one.
> For example, in the CIFAR-100 experiment, we have a $5\times 100$ matrix and in the ImageNet experiments, we use a $5\times m$ matrices where $10 \leq m \leq 40$.
>
> **At first, it appears that the distribution would be a sample. However, fixed distribution is used for a set of experiments. For example, it is either [0.5, 0, 0, 0, 0.5] or [0.2, 0.2, 0.2, 0.2, 0.2] for the entire experiment. Hence, presenting it as "sampled" is confusing. The best results come when you have the top1 and the sum of the top five values. Hence, the initial discussion and intuition can be improved a bit.**
> We are not sure what you mean exactly with the distribution being a sample. We do not sample the distribution, but instead we model sampling *from* a distribution.
> Could you point us to the line from where you understood that we "sample the distribution" such that we can rework it to avoid this confusion or ambiguity in the revision?
> Note that, when $k$ is drawn from the distribution parameterized by $[0.5,  0,  0,  0,  0.5]$, this is done sampling-free as we can compute the expectation value in closed form, which makes learning effective and efficient.
>
> **The improvements on Noisy Student EfficientNet-L2 are negligible. 88.35 to 88.36 is certainly not statistically significant. Were experiments for Table1 were also ran 10 times (like table 2?).**
> Originally, we ran the experiment only once. However, now, we have run the experiment 5 times. Among the 5 runs, the average accuracy with DiffSortNets is not $88.36\%$ but instead $88.384\% \pm0.009 \%$.
> Note that “88.35” is a reported accuracy.
> When loading the public model in PyTorch, it yielded an accuracy of $88.33\%$, and this is what should be used for a fair comparison because it is also the exact model that we base the results for our method on.
> Using a t-test (with significance level of 0.01), we find that our model is significantly better regardless of which of the two baselines one compares to: Comparing to the reported $88.35\%$, our method is significantly better (p value $=0.00074$) and comparing to the $88.33\%$ in the PyTorch implementation, our method is also significantly better (p value $=0.00012$).
> We will address this in the revision and clarify the significance of the improvement.
>
> **Please mention the number of rows that were reconstructed for each experiment. The number of columns (m) is mentioned in the experiments but not the number of rows.**
> The number of computed rows is always equal to the maximum possible $k$, i.e., in the main experiments it is 5 rows. (Except for Sinkhorn where all $m$ rows need to be computed.) In any case, only the top 5 rows are used for the loss function as the weights for the other rows are 0.
>
> **Berrada et al. was used to train the model from scratch. It would be worth comparing their loss for fine-tuning purposes as well. I think that would be a fairer comparison.**:
> In our work, the numbers for the method by Berrada et al. (Smooth top-k loss in Tables 2, 3, and 4) are always in the same setting as for our method, i.e., for CIFAR-100 training from scratch and for ImageNet fine-tuning. For fine-tuning, we use the same embeddings for both our method and for the Smooth top-k loss. We will clarify this in the revision.
>
> We are looking forward to hearing back from you.

---

> > ### Comment · Reviewer_RKLW · 2021-11-19
> > **The rebuttal addresses most of my concerns.**
> >
> > What I meant on the second point, was the same as Abaq, that P_K appears more like a set of weights, rather than a probability distribution. The initial feeling I got was that P_K will vary at every iteration (drawing from a distribution).
> >
> > Another minor clarification needed: 16<m<50 or 10<m<40?
> >
> > Overall, most of the concerns raised by me are addressed. I am increasing my rating for the paper.

---

> > > ### Author Response · Authors · 2021-11-19
> > > **Thank you for your response!**
> > >
> > > We really appreciate your response and raising your score!
> > >
> > > We have just uploaded a revision.
> > >
> > > We finally finished running the experiments for Table 1 10 times now to make it conform to the other tables.
> > > We have included the t-test that shows statistical significance in the revision (for both top-1 and top-5 acc.).
> > >
> > > We also addressed the other aspects that we promised to address.
> > >
> > > **What I meant on the second point, was the same as Abaq, that P_K appears more like a set of weights, rather than a probability distribution. The initial feeling I got was that P_K will vary at every iteration (drawing from a distribution).**:
> > > We hope to have clarified this in the revision by adding a note in the introduction. If you think additional clarification is helpful, we are happy to add it.
> > >
> > > **Another minor clarification needed: 16<m<50 or 10<m<40?**
> > > In the main experiments $m\in\{16, 50\}$ (16 for ImageNet-1K, 50 for ImageNet-21K-P), and in Figure 3 where we study the effect of $m$ in the ImageNet-1K setting $10<m<40$.
> > >
> > > If you have any additional questions or concerns, don't hesitate to ask.
> > > We are looking forward to hearing back from you.

---

### Official Review · Reviewer_Abaq · 2021-11-02

**Correctness:** 3
**Technical Novelty And Significance:** 3
**Empirical Novelty And Significance:** 3
**Recommendation:** 6
**Confidence:** 2

**Main Review:**

strengths
1. The idea of using different probability distributions for k is interesting. The results also demonstrate the effectiveness of this idea.
2. The experiments of incorporating different sorting methods are comprehensive.


weaknesses
1. In my opinion, the P_K is more like a set of weights, rather than a probability distribution. If it is the case, I recommend improving the descriptions to reduce confusion.
2. It would be nice to present an experiment with conditional probability distributions for k of different classes based on their semantic meaning (like person, animal). I think it is also a significant contribution of this paper.


**Summary Of The Paper:**

This paper addresses Top-k classification learning. Based on the recent progress on differentiable sorting and ranking, the author proposes a loss function for top-k classification where the k is not fixed but follows a given probability distribution. To improve the efficiency, a splitter selection network is proposed so that fewer layers are required for the sorting network.

The proposed loss function can be combined with different sorting methods. In experiments, the loss function is shown to be effective in training a model from scratch on Cifar10. It can also be used in fine-tuning on ImageNet dataset and has performance gain.

**Summary Of The Review:**

This paper proposes a flexible loss function for top-k classification, providing useful insights for image classification. So it is worth of reading for the researchers in this area.

---

> ### Author Response · Authors · 2021-11-10
> **Response to Reviewer Abaq**
>
> Thank you very much for your encouraging review.
> Please let us know whether you have further suggestions and questions.
>
> **P_K is more like a set of weights**:
> We agree that they can also be referred to as weights.
> We refer to it as a probability distribution because, from a theoretical point of view, we model it by assuming $k$ is unknown and drawn from $P_K$.
> In this context, we consider weights (summing up to one) and (categorical) probability distribution to be synonymous.
> In the revision, we clarify that the cumulative sum of $P_K$ is a list of weights for aggregating the rows of $P$, which yields $p$.
>
> **It would be nice to present an experiment with conditional probability distributions for k of different classes based on their semantic meaning (like person, animal). I think it is also a significant contribution of this paper.**
> We agree that this would be an interesting addition, and we had originally considered adding it to the paper; however, as the space is limited and one experimental result already had to be deferred to the supplementary material, we decided to focus on the presented experiments.
>
> We are looking forward to hearing back from you.

---

> > ### Comment · Reviewer_Abaq · 2021-11-29
> > **Additional comments**
> >
> > I agree with the authors that P_K can be considered as a probability distribution because it is non-negative and summed to 1. However, if P_K is a probability distribution, I would always expect more probabilistic motivation for the loss function (Eqn. 2). For example, I would expect Eqn. 2 to be derived by minimizing one KL divergence between a label matrix (generated by P_K) and permutation matrices P, rather than a weighted sum of multiple probabilities.
> >
> > Additional comments to the "response to reviewer RKLW".
> > I don't think the significance tests convince me on the question of "The improvements on Noisy Student EfficientNet-L2 are negligible".
> >
> > You have shown that your model is better than the baseline. However, the improvement is from 88.33 to 88.37. I would still believe that the improvement of 0.04% is negligible, which couldn't demonstrate the usefulness of the proposed method.

---

> > > ### Author Response · Authors · 2021-11-30
> > > **Response to additional comments**
> > >
> > > Thank you very much for your response!
> > > If you have any further questions, please don’t hesitate to ask.
> > >
> > > **I agree with the authors that P_K can be considered as a probability distribution because it is non-negative and summed to 1. However, if P_K is a probability distribution, I would always expect more probabilistic motivation for the loss function (Eqn. 2). For example, I would expect Eqn. 2 to be derived by minimizing one KL divergence between a label matrix (generated by P_K) and permutation matrices P, rather than a weighted sum of multiple probabilities.**
> > >
> > > Equation 2 is derived from Equation 1, which is the probabilistic basis for it. Note the expectation value over $k\sim P_K$ in Equation 1. The weighted sum in Equation 2 is the result of the expectation value from Equation 1.
> > >
> > > We are not sure how a label matrix could be generated by $P_K$: For each image, only 1 label is given. A label matrix would require a known ranking of multiple classes, which is not available, and if it was available, this would be a different problem setting. $P_K$ also does not include information for computing a label matrix.
> > >
> > > **Additional comments to the "response to reviewer RKLW". I don't think the significance tests convince me on the question of "The improvements on Noisy Student EfficientNet-L2 are negligible”.**
> > >
> > > **You have shown that your model is better than the baseline. However, the improvement is from 88.33 to 88.37. I would still believe that the improvement of 0.04% is negligible, which couldn't demonstrate the usefulness of the proposed method.**
> > >
> > > We understand that an absolute improvement of 0.04% can seem small.
> > > However, we would like to point out that we only fine-tune the last layer of an existing network. This is many orders of magnitude less expensive than retraining. For example, while our method may take a few hours on a single CPU thread or minutes on a GPU, the original training of the model required 6 days on a Cloud TPU v3 pod (!). A Cloud TPU v3 pod has 32 TB of HBM memory, 2048 cores, and corresponds to a cluster of the order of around 1000 GPUs (100 petaflops in the TPU v3 pod vs 120 teraflops on a V100 GPU; the TPU v3 pod is generally better connected than 1000 GPUs would be).
> > > Training the model once would cost based on Googles pricing 384x4x24x6=221 184 USD.
> > > Using a single GPU, we expect the runtime of the original training to be 16.4 years.
> > > Remind, our method takes minutes on a GPU (while the GPU is not even fully utilized).
> > > The extreme contrast of training costs (a factor of around 100 000 — 500 000) means that with fine-tuning with our method, we get a much better return-on-investment.
> > > In our opinion, even an absolute 0.01% improvement (given that it is significant) at 0.0002% to 0.001% of the training cost is worth it.
> > > We also note that our improvement corresponds to a 0.34% relative improvement.
> > >
> > > We will include this discussion in the final version.
> > >
> > > In other settings, we also achieved larger improvements, e.g., a ~10% relative improvement from 97.80 to 98.00 (top-5 acc for the ResNeXt-101 32x48d WSL backbone). This improvement is larger than the improvement from increasing the model size by a factor of 1.5 (from 32x32d to 32x48d, see Figure 2).

---

> > > > ### Comment · Reviewer_Abaq · 2021-12-03
> > > > **Thanks for your explanation. But it does not 100% convince me.**
> > > >
> > > > I have read your response.
> > > >
> > > > It does not make a lot of sense to compare the "return-on-investment" between a fine-tuning model and a training-from-scratch model.
> > > >
> > > > From my perspective, an improvement of 0.04% is still very small. I am looking forward to the opinions from other reviewers and AC.
> > > >
> > > > I think it would be useful to report the standard deviations for 10 evaluations in table 1.

---

> > > > > ### Author Response · Authors · 2021-12-03
> > > > > **Response.**
> > > > >
> > > > > Thank you very much for your response!
> > > > >
> > > > > We would like to emphasize that the 0.04% absolute improvement is only one out of many results. We think that the value of the paper is that it proposes a new and efficient way to train networks, and demonstrate a constant improvement on various tasks, datasets, and backbones.
> > > > >
> > > > > For Noisy Student EfficientNet-L2 Top-k DiffSortNets the exact mean and standard deviations are 88.374±0.016 | 98.678±0.013. We propose to include standard deviations for Table 1 in the final version.
> > > > >
> > > > > If you have any further questions, please don’t hesitate to ask.

---

### Official Review · Reviewer_575y · 2021-11-03

**Correctness:** 3
**Technical Novelty And Significance:** 3
**Empirical Novelty And Significance:** 3
**Recommendation:** 3
**Confidence:** 4

**Main Review:**

Strengths:
	The motivation of this paper is clear to draw k from a probability distribution for training.
	The idea of this paper is pretty novel and exciting which makes the classification model robust.
	The extensive experiments conducted on five data sets are sufficient to show the advantages of the proposed idea
Weaknesses:
	The details of the differentiable sorting networks is not represented. How to rank the predicted scores of the final classification layer and get the probability distribution?
	In figure 1, the first row (rank1) are multiplied by 1 and the second row(rank2) are multiplied by 0.5. Please explain the reason.


**Summary Of The Paper:**

This paper proposes a loss to relax the assumption of using a fixed k for top-k classification learning. The authors use the existing differentiable sorting and ranking operators. Experimental results also achieve a state-of-the-art on ImageNet.

**Summary Of The Review:**

This paper derives a family of top-k cross entropy losses which is a novel practice. The experimental analysis on ImageNet including the impact of the distribution and ranking set size m, etc, is concrete and sufficient.

---

> ### Author Response · Authors · 2021-11-10
> **Response to Reviewer 575y**
>
> Thank you very much for your encouraging and positive review.
>
> **The details of the differentiable sorting networks is not represented. How to rank the predicted scores of the final classification layer and get the probability distribution?**:
>
> We extended the description of differentiable sorting networks.
> We would like to point out that we also use other differentiable sorting methods and therefore want to present each of these methods with equal emphasis.
>
> As to how to obtain the probability distribution: using differentiable sorting networks (or any other differentiable sorting method like Sinkhorn) we sort the class scores.
> These methods produce a differentiable permutation matrix P, which yields a sorted vector when P is multiplied by the input vector.
> For this, differentiable sorting networks have a matrix for each layer which are combined into P by multiplication.
> In the Sinkhorn sort algorithm, P is obtained as the resulting transport matrix of the Sinkhorn iteration.
> This matrix P is then a doubly-stochastic matrix which means each row and each column is a probability distribution.
>
> **In figure 1, the first row (rank1) are multiplied by 1 and the second row(rank2) are multiplied by 0.5. Please explain the reason.**:
> In the example we use a 50% top-1 and 50% top-2 loss.
> For the top-1 component, we only consider rank1 and therefore already give it a weight of 0.5.
> For the top-2 component, we consider both rank1 and rank2 (because both rank1 and rank2 are valid ranks for top-2, i.e., for among the top 2 ranks).
> Therefore, we add 0.5 to the weights for rank1 and for rank2, which is why we obtain a total weight of 1 for rank1 and a total weight of 0.5 for rank2.
> In fact, rank1 always has a weight of 1 because if an element is at rank1 (i.e., it is the maximum) it is among the top-k for any choice of k and the sum of probabilities in $P_K$ has to be 1.
> We extended the description in the revision to clarify this reasoning.
>
> Could you please explain your reasons for your recommendation “reject” and what we could do to improve our work, considering that your review seems throughout very positive?
>
> We are looking forward to hearing back from you.

---

> ### Author Response · Authors · 2021-11-22
> **Response to Reviewer 575y (2)**
>
> Considering your positive review, we would really appreciate if you could please consider raising your score or explain your reasons for your recommendation “reject”.
>
> Thank you.

---

### Author Response · Authors · 2021-11-19
**Revision of the draft.**

We really appreciate all four reviewers for their valuable comments.

We have uploaded a revised draft incorporating the reviewer feedback, where we have colored our main changes in blue.

We really hope our responses and revisions address all reviewers’ concerns!

---

### Decision · Program_Chairs · 2022-01-20

**Decision:**

Reject

**Comment:**

The main consensus among the reviewers was that although the approach is interesting, this submission suffers from two main weaknesses:

- The methodology is not very novel, and the proposed parts of the method not well justified (in particular regarding the interplay of a differentiable sorting approach and of the random choice of k)

- The results, compared to a standard cross-entropy loss are not very convincing: there does not seem to be a statistically significant advantage.